# A Top-Down Digital Mapping of Spatial-Temporal Energy Use for Municipality-Owned Buildings: A Case Study in Borlänge, Sweden

**Samer Quintana** [1,2,*] **, Pei Huang** [1] **, Mengjie Han** [1] **and Xingxing Zhang** [1]

1   School of Technology and Business Studies, Dalarna University, 79188 Falun, Sweden; phn@du.se (P.H.);
    mea@du.se (M.H.); xza@du.se (X.Z.)
2   Department of Engineering Sciences, Uppsala University, 75237 Uppsala, Sweden
*   Correspondence: ssq@du.se; Tel.: +46-7309-48879

**Abstract:** Urban energy mapping plays a crucial role in benchmarking the energy performance of buildings for many stakeholders. This study examined a set of buildings in the city of Borlänge, Sweden, owned by the municipality. The aim was to present a digital spatial map of both electricity use and district heating demand in the spatial–temporal dimension. A toolkit for top-down data processing and analysis was considered based on the energy performance database of municipality-owned buildings. The data were initially cleaned, transformed and geocoded using custom scripts and an application program interface (API) for OpenStreetMap and Google Maps. The dataset consisted of 228 and 105 geocoded addresses for, respectively, electricity and district heating monthly consumption for the year 2018. A number of extra parameters were manually incorporated to this data, i.e., the total floor area, the building year of construction and occupancy ratio. The electricity use and heating demand in the building samples were about 24.47 kWh/m$^2$ and 268.78 kWh/m$^2$, respectively, for which great potential for saving heating energy was observed. Compared to the electricity use, the district heating showed a more homogenous pattern following the changes of the seasons. The digital mapping revealed a spatial representation of identifiable hotspots for electricity uses in high-occupancy/density areas and for district heating needs in districts with buildings mostly constructed before 1980. These results provide a comprehensive means of understanding the existing energy distributions for stakeholders and energy advisors. They also facilitate strategy geared towards future energy planning in the city, such as energy benchmarking policies.

**Keywords:** digital mapping; spatial; temporal; energy use

## 1. Introduction

Buildings represent large energy end-users worldwide. In the EU, buildings currently consume over 40% of total primary energy usage [1]. With its sights set on the new paradigm shift regarding energy production, efficiency and climate change, Sweden will implement strategies to reach national targets for energy efficiency in the building sector by 2050. According to these targets, energy use per square metre should decrease by 20% by 2020 and 50% by 2050, in comparison with use in 1995—this is a national target for energy efficiency in the housing sector [2]. In 2010, over 50% of the world's population were living in urban areas. By 2050, this number is expected to reach 75% [3]. Urban development and the expansion of cities, through the modification of land uses (from natural to artificial), cause a shift in the local energy budget and energy supply/demand patterns. Such a transformation has significantly changed the microenvironment and the related energy usage in urban cities [4]. The mapping of urban building energy plays a crucial role in understanding the multitude of agents that take part in the energy performance of buildings and thus in setting up the benchmarks in different districts for various stakeholders.

In Swan and Ugursal's study, the modelling approaches for energy consumption in a number of buildings were classified into bottom-up or top-down approaches [5]. The bottom-up approach is more appropriate when there is a need to evaluate the energy consumption based on a highly detailed level of data and to model technological systems [6]. Bottom-up models can be divided into two types: deterministic (or engineering) and statistical. The statistical methods search for correlations, utilizing a sample of information from energy bills as a source of data for energy modelling and analysing the link between energy consumption and a range of different variables (e.g., building shape, age and occupant behaviour) [7]. They can also take into account socioeconomic effects in the equations. They calculate reliable consumption based on the available information on the current status of buildings. However, due to their strong dependency on available historical consumption data, these bottom-up statistical methods are restricted to predicting the impact of new technology options and energy saving potential after the application of refurbishment measures [8]. The bottom-up deterministic methods are detailed models which are based on thermodynamic relationships and heat transfer calculations [9]. The main advantage of an engineering-based method is the ability to predict energy saving potential for buildings when some renovation measures are to be implemented [10]. These modelling approaches require a large amount of information about the building structures and parametric input to estimate the energy usage of a set of reference buildings of the stock based on a numerical model. Additionally, the evaluation of urban planning scenarios is computationally extensive, and the availability of construction and geometrical data needed as input for the models is very scarce. The top-down approaches treat the entire residential sector as one energy sink. Unlike the bottom-up approaches, the top-down methods are suitable for a large-scale analysis and not for the identification of the possible improvements to the building at urban and local levels [11]. Compared with the bottom up-approaches, the top-down methods are relatively easy to develop based on the limited information provided by macroeconomic indicators, such as price and income, technology development pace and climate. As summarised by Swan and Ugursal, the top-down approaches have advantages including the capacity for long-term forecasting in the absence of any discontinuity, inclusion of macroeconomic and socioeconomic effects, the simple input information required and the capacity to encompass trends [5].

Both the bottom-up and top-down approaches can assist the spatial–temporal analysis of the energy demand at the district level. For instance, Schneider et al. (2017) developed two bottom-up statistical extrapolation models for spatial–temporal analysis of the geo-dependent heat and electricity demand of a building stock located in Switzerland [12]. They calculated the heat demand using a statistical bottom-up model applied at the building level. Due to the large variability in the electricity usage, they estimated the municipality-level electricity load curve by combining socio-economic indicators with the average consumption per activity and/or electric device. Chen et al. (2019) established a Geographic Information System (GIS) based multi-criteria index system for spatial–temporal analysis of the energy demand in a university located in China [13]. They used the developed system to investigate the characteristics of (i) the temporal dynamic, (ii) the load fluctuation and (iii) the district load spatial distribution as well as the coupling relationships of power loads for heating/cooling between single buildings and the entire university district. They also implemented principal component analysis to identify the buildings which had large impacts on the district power demand. Unlike most of the existing approaches, which estimate the district energy demand at different spatial–temporal levels as functions of the characteristics of either individual buildings or cities and their occupancy levels, Mohammadi and Taylor (2017) connected spatial–temporal heterogeneous human behaviour with the city-level building energy use [14]. They first examined the temporal manifestation of the energy use fluctuations in urban buildings driven by spatial mobility patterns of the population, and then they developed a multivariate auto-regressive model for spatial–temporal analysis of the urban-level building energy demand in the City of Chicago based on a yearly individual positional record. Their study reveals that human

mobility can account for the collective energy consumption in urban spaces. As can be seen from the abovementioned literature, spatial–temporal analysis of the geo-dependent urban-level heat and electricity demand is important for urban-scale planning and can bring several benefits: (i) it is beneficial for the construction of a geo-referred database for a specific location; (ii) it enables the estimation of the energy saving potentials that can be achieved by different retrofit programs and thus assists decision making; (iii) it supports the investigation of the influential factors affecting electricity demands and the optimization of the operation and management of district heating/cooling systems and district power dispatches.

Besides the top-down and bottom-up approaches, there is also a typology approach, which is based on the synthetic characteristics of a group of buildings. The European TABULA project defined building typology as "a systematic description of the criteria for the definition of typical buildings as well as a set of exemplary buildings representing the building types" [15]. It takes into account aspects such as climate, period of construction, spatial and housing models, technologies, design rules, building codes, planning regulations, economic constraints, building construction techniques, the organization of construction companies and worksite organizations [16]. Dascalaki et al. has demonstrated that the typology approach is effective in investigating the energy performances of building stocks [17]. By drawing on various building typologies, an energy benchmarking system could be developed as representative of a large percentage of the entire urban building stock. This approach has also been utilised in European Commission energy projects like RePublic_ZEB [18]. An effective tool to support the typology approach in analysing urban building stock patterns and "typologies" is spatial cluster analysis. For instance, Lucchi et al. conducted a spatial cluster analysis using data-mining methods (i.e., an hdb-scan algorithm) and a GIS method to investigate the energy performances in a historic town in Calavino, Italy [19]. Such clustering analysis can overcome the inaccuracies related to the application of the traditional building stock analysis approach. Similarly, Miao et al. proposed a clustering method to automatically extract and identify urban spatial patterns and functional zones based on massive amounts of volunteered geographic information collected in Beijing, China [20]. The study results show that these methods can effectively identify urban spatial patterns and thus can contribute to urban energy simulation.

In the context of sustainable cities, spatial visualization is a very effective approach that can help decision-makers in the urban planning process create future energy transition strategies and implement energy efficiency and renewable energy technologies. The most fundamental energy visualization tools use simple lines, pie charts and bar charts to show the energy usage patterns over time at the individual building level. For instance, the Pulse Dashboard presents trend-line energy consumption data for each commercial building [21]. The Building Dashboard presents energy usage using bars [22]. Other 2D visualization techniques include cluster maps, component planes, spiral displays, time logs and thematic 2D maps [23,24]. However, as the number of analysed buildings increases, these conventional visualization techniques may not perform well due to the limited information that can be presented; most of them can only reveal the temporal characteristics of land but not the spatial characteristics. Compared with 2D visualization, 3D visualization is more realistic and psychologically appealing for the human brain. Geographic information system (GIS) techniques can be used for the visualisation of the energy demand or production in buildings from the urban to the regional scale, or even at a national one. These visualization techniques include "hit maps" (i.e., aggregated data in 3D charts) [24] and 3D city models with semantic objects [25]. There are many studies using GIS techniques to visualise the energy data in building stocks. For instance, Mattinen et al. (2014) developed a method for estimating and visualizing the energy use and greenhouse gas emissions from a residential building stock located in the Kaukajärvi district, Finland [26]. Using such a visualization model, they also analysed the impacts of behavioural and technical changes on the energy performance in the building stock. Finney et al. made a comprehensive mapping of heat sources and sinks in Sheffield City,

UK [27]. Based on the heat source mapping, they linked these smaller systems to create a combined heat and power-based urban-scale network of energy generation and delivery. Huang et al. (2019) used a GIS technique to obtain the roof area in Kowloon district in Hong Kong. Using the obtained roof area, they evaluated the solar power potential that would be available for the whole district by installing rooftop PV panels, which was then used as the input for designing public charging stations. The solar PV potentials were visualised using different colours on the Kowloon district map [28]. Similarly, Ramachandra and Shruthi used the GIS technique to map the wind energy resources of Karnataka state, India. Based on the wind-power mapping, they analysed the variability of these resources, considering spatial and seasonal aspects [29]. Despite the abovementioned literature, until now the utilization of 3D visualization in spatial–temporal analysis of urban-scale energy usage has been very limited.

Although there are existing studies of mapping energy uses in different cities, spatial energy analyses in local municipalities are necessary as they will be different in various city and culture contexts. Specific consideration should be paid to the differences between cities when the aim is to optimise the integration of urban energy systems operated in buildings and promote renovation and renewable energy systems. This is because cities differ from each other at the local, national and international levels from the perspectives of geography, socio-economy, culture, infrastructure, and information platform. The types of cities and districts determine the kinds of users and needs and consequently the nature (qualitative and quantitative) of the policy/regulation schemes and the calibration/adjustment of the energy infrastructures. Citizens' behaviours and needs/preferences with regard to energy may be different from each other in different cities, which can lead to great differences in energy demand. Within the same framework of transforming to a sustainable and liveable city, different areas must not only adopt standardised approaches but also take into account specificities at the local level. Dedicated research into cities and districts at the local scale is therefore of paramount importance to ensure the proper mix between international/national scenarios and local measures.

The urban energy mapping and analysis for Borlänge city have not yet been done. This study therefore aimed to cover this research gap by examining a set of buildings owned by the municipality of Borlänge, Sweden. The first step of the study was to conduct a spatial–temporal analysis of both electricity use and district heating demand. A top-down approach was considered based on the energy consumption data of the municipality-owned buildings. It was expected that this study would be able to provide insights that allow an understanding of the existing local energy distributions. It also facilitates strategy geared towards future energy planning in this city.

This paper is structured as follows: Section 2 indicates the data sources and the methodology used to process the data; results and discussion are presented in Sections 3 and 4; a conclusion is included afterwards.

## 2. Data Sources and Research Methods

### 2.1. Data Sources

Acquiring the necessary data to create an urban model can be a difficult endeavour. New general data protection regulation (GDPR) laws instituted by the European Parliament regulate how data can be acquired, handled and stored in order to protect the privacy of individuals [30]. Energy consumption data include sensitive information that falls within the bounds of the new regulation, greatly complicating the data acquisition. Depending on the data resolution, storing the information can be complicated as it may not be kept for long periods of time or may be stored in obsoletes systems, making it difficult to be of use.

The primary source of the data used for this model was Tunabyggen, a municipality-owned company that constructs, manages and rents a set of buildings in the Borlänge municipality. The data were provided in the PDF format, with a total number of 375 pages of monthly data for electricity demand, district heating and hot water flow rate for the year 2018. The geographical information, specifically the vector data for the property

information and LIDAR data for the Borlänge municipality, was obtained from the official Swedish surveying institution, Lantmäteriet. Other social statistics and specific data such as building year of construction, percentage of occupation, demographics and typologies were acquired from hitta.se, a Swedish search engine that offers a telephone directory, addresses and maps. To complete and validate the model, it was necessary to use some extra information that was obtained by visual inspection, including the number of floors and the area and shape of the roofs. The flowchart in Figure 1 further describes the processes, databases and validation operations.

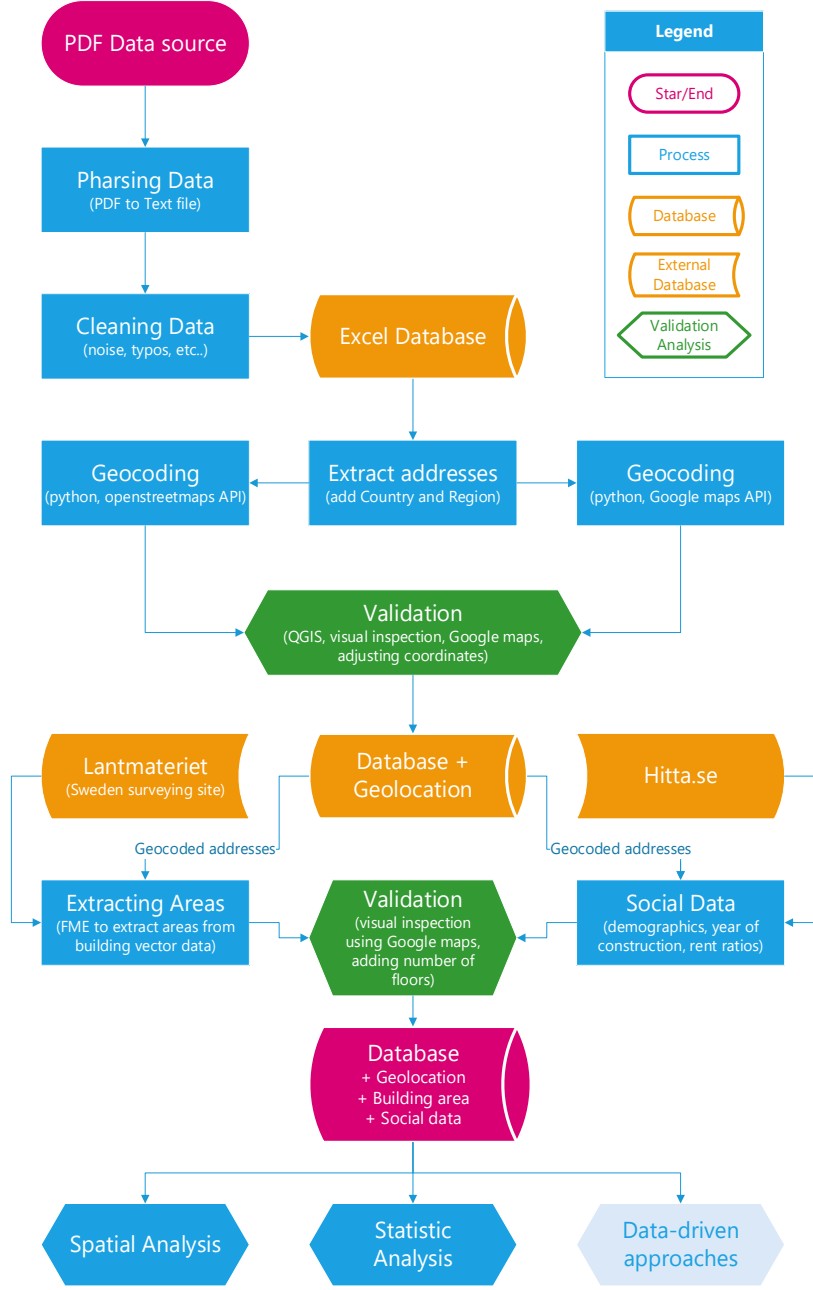

**Figure 1.** Flowchart for data processing, extraction, geocoding and validation.

### 2.2. Data Extraction

The first step in the process was to extract the information from the data source provided. The archaic PDF data structure format had to be transformed into a common format that could be used by other applications. In order to extract the data, a custom

Python script was written to parse out the information. Then, the data were further inspected for missing data and error correction. From the 375 pages in PDF format, a total of 262 addresses and 463 entries of monthly data for electricity (kWh), district heating (MWh) and flow rate (m3) for the year 2018 were extracted.

*2.3. Geocoding*

The addresses extracted from the data source were further expanded to the city and the country. Then, they were processed with a Python script using an application program interface (API) for OpenStreetMap (OSM). Figure 2 shows the script flowchart that was run, which used the pandas and geopy libraries. In parallel, another script was used to connect to the API geocoding services of Google Maps. Two outputs from each geocoding service were obtained with the longitudes and latitudes of the addresses. The output format for the coordinate system was the standard LL-WGS84 [31]. The locations for a total of 222 out of the 262 entry points were found on the first iteration.

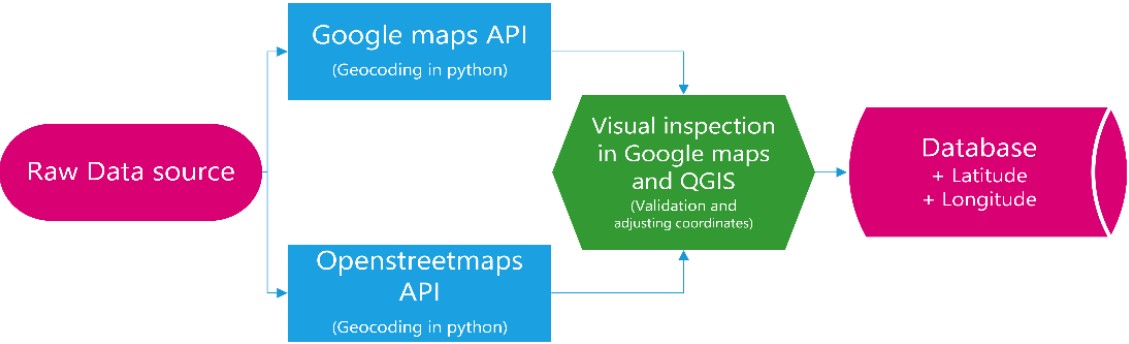

**Figure 2.** Flowchart for OSM and the Google Maps Python application program interface (API) geocoder.

*2.4. Geocoding Validation*

The results were plotted and further inspected for validation. During this process, the locations were geocoded and manually centred in the property area, as shown in Figure 3. The red dots were the geocoded locations that were manually centred in the building properties (green polygons). The output amounted to 238 out of the 262 total addresses, leaving a total of 24 addresses and 31 entry points that, due to unspecific naming, we were not able to geocode until manual visual inspection and analysis of the context were undertaken. The preliminary result generated a total of 250 geocoded addresses and 12 unclarified ones.

*2.5. Area Merger Code, Area Validation*

The next parameters were extracted from the Swedish survey database Lantmäteriet [32]. The building property vector information was provided in a shapefile (.shp) format, a digital vector storage format for storing geometric location and associated attribute information.

Using the Feature Manipulation Engine (FME) tool, it was possible to extract and calculate the areas for the geocoded address points [33]. This information was compared to the visually inspected area in order to analyse its accuracy. The extra information stored in the shapefiles was incorporated into the dataset. This information included a building description, coordinates in the Swedish reference system SWEREF-99-TM and a unique object identity [31].

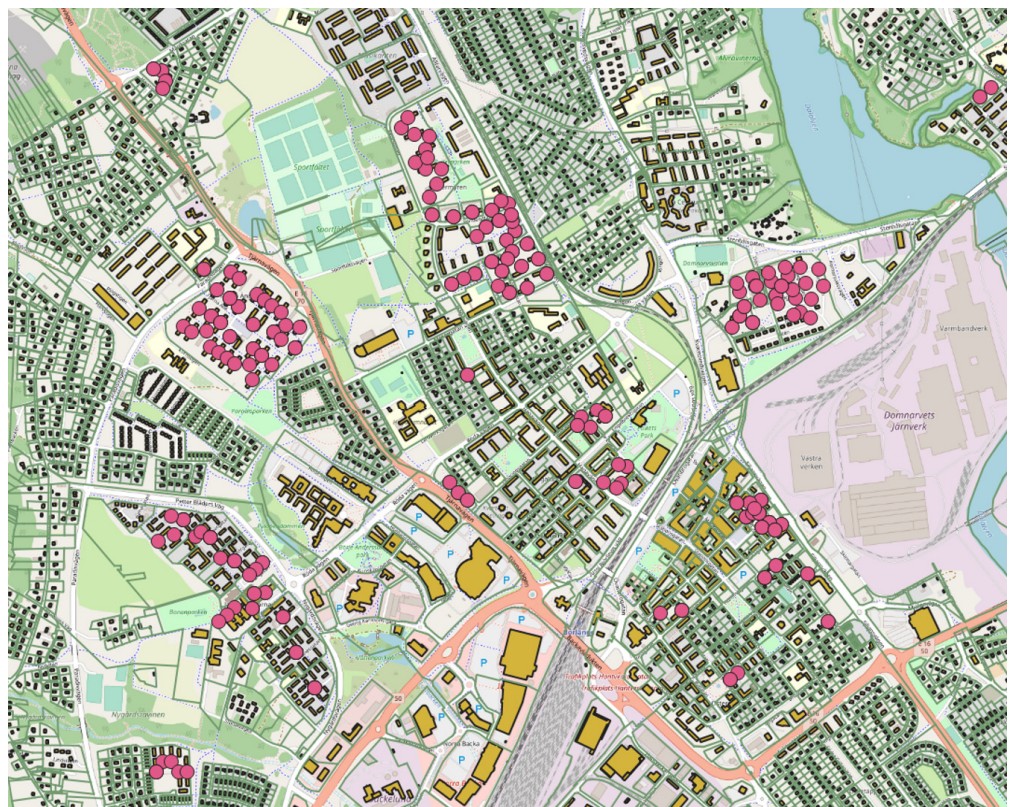

**Figure 3.** Geodata: building property vectors with adjusted coordinates.

### 2.6. Data Processing

All the different sources of information were finally combined together and inspected for errors or inconsistencies. The total building area was calculated using the number of floors and the buildings' polygon surface areas. Finally, the results for the energy consumption, electricity and district heating in kWh/m² for the year 2018 were obtained. From the 250 total addresses that were geocoded, 28 addresses were excluded from the analysed dataset due to missing, erroneous or abnormal information. The initial dataset contained 236 entries for electricity demand and 108 for district heating demand, which were reduced to 228 and 105, respectively, due to the following reasons: (i) some entries related to utility building samples that had no coherent energy demand on a normalised per metre square surface—for example, the energy demands of some laundry buildings were not representative for this dataset as they were detached from the buildings they provided services to; (ii) the building occupancy ratios for the entries were close to zero—some of the buildings in the sample were unoccupied, so their energy demand was close to zero. The final sample dataset consisted of 228 buildings for the electricity data and 105 buildings for the district heating data.

## 3. Results

### 3.1. Statistic Data Analysis

In the considered building samples, all of the buildings were residential buildings and related facility buildings (such as laundries, storage, etc.). The energy use was normalised by dividing it by the heated floor area. The definition of the heated or living floor area has a large impact on the magnitude of the area-specific energy requirement. In Sweden, the heated floor area is defined as the floor area that is heated to more than 10 °C. As a result, in this study, we assumed the heated floor area was on average 87% of the total external floor area for the analysis [34]. In addition, electricity demand was further normalised by considering the occupancy ratio of each building. For heating demand there was no need

to consider the occupation ratio, as it is common in Sweden for heating systems to stay on even when a building is unoccupied.

The data were normalised by dividing the total energy consumption by the total living space area. This worked for most cases, i.e., self-sufficient homes and buildings with integrated facilities. In other cases, when the utility facility was in a detached building, it was unclear how many buildings it provided services to. Therefore, the area of the facility in itself was not representative of the total area it served, giving an impression of a very energy-inefficient building. In Sweden, most communal arrangements have a dedicated laundry, garage or storage facility for the community. The utilities room is usually in the basement of apartment buildings, providing an assortment of laundry and/or ironing machinery. In other cases, this facility can be completely detached from the main building. The annual electricity demand for lighting and appliances in the building samples is shown in Figure 4.

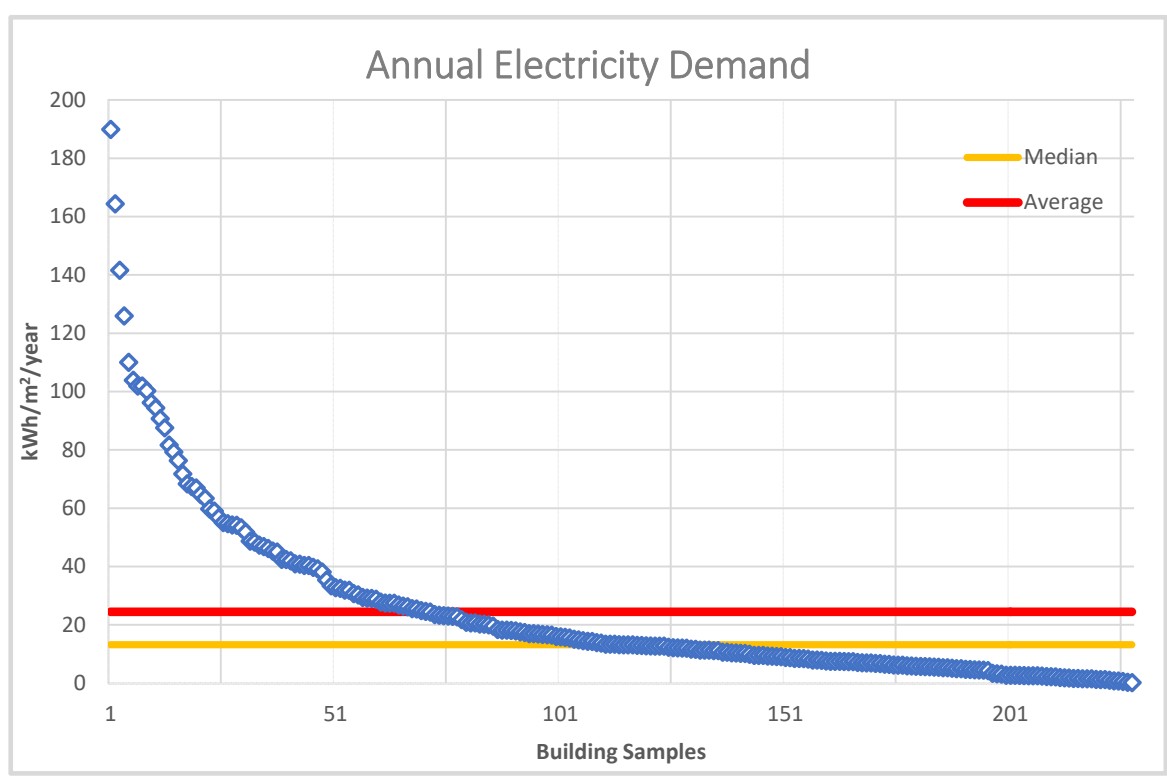

**Figure 4.** Annual electricity demand for building samples.

The average electricity demand of the 228 building samples was 24.47 kWh/m$^2$, with a total range from a minimum of 0.16 kWh/m$^2$ to a maximum of 189.89 kWh/m$^2$. Compared to the average electricity demand of 30–36 kWh/m$^2$ in the Swedish context [35], the average electricity demand of the building samples was reasonably low. This corresponded with the build year, zone, occupant background and purpose of the buildings, as most of the occupants in the sampled buildings were life renters, students or had a relatively low income. The median electricity demand was 13.17 kWh/m$^2$, which means that 50% of the sampled buildings demanded less electricity than this value. Furthermore, over 75% of the sampled buildings achieved electricity use lower than 30 kWh/m$^2$.

The Swedish Housing Agency's building rules [36] stipulate requirements for the energy performances of buildings depending on their use, end-use heating system and climate zone. The energy performance (heating demand) requirements are given as the specific energy use, comprising the purchased energy for space heating, domestic hot water

and electricity for fans and pumps but excluding electricity for household appliances and lighting [37]. The annual heating demand for the building samples is displayed in Figure 5.

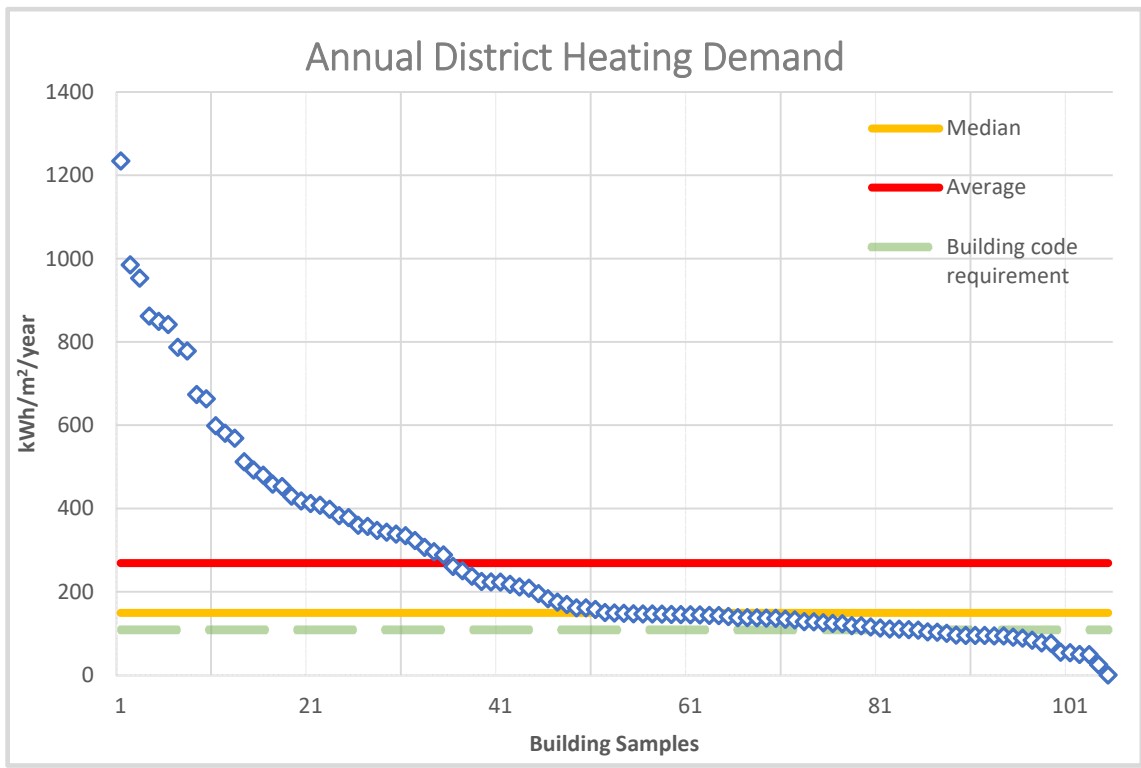

**Figure 5.** Annual heating demand for building samples.

The average heating demand of the 105 building samples was 268.78 kWh/m$^2$, with a total range from a minimum of 0.41 kWh/m$^2$ to a maximum of 1234.52 kWh/m$^2$. Borlänge city belongs to Climatic Zone II in Sweden, for which the new building code requires an annual energy use of up to 110 kWh/m$^2$ for non-electrically heated buildings (i.e., heated with district heating). In addition, the criteria for passive houses include even have higher requirements, with a value up to 35% lower compared to the building code [38]. Thus, the average heating demand in the building samples was much higher than either the building code or the passive house standard, about twice of requirement stipulated by the building code and three times the requirement of the passive house standard. The median heating demand was 149.34 kWh/m$^2$, which means that 50% of the building samples demanded less heating than this value. Approximately 21% of the building samples achieved a lower heating demand than 110 kWh/m$^2$. The difference between the different municipalities is clear. In Gävleborg, it was found that the average heating demand was about 185 kWh/m$^2$ in 2010. Across the whole of Sweden, the average annual energy use for heating in one- or two-dwelling buildings was reported to be about 158 kWh/m$^2$ per year in 2014 [39]. Therefore, the heating use in Borlänge city was found to be at a high level when compared to that of the closest regions and the average figure for the country. However, this high energy demand can be explained by the fact that over 56% of the buildings in the sample were constructed before 1980 and therefore may not be energy-efficient dwellings.

Some of the building samples with a high energy demand corresponded to small districts or clusters of buildings. Even though the total area was aggregated and normalised, it is possible that some heating surfaces for common and utility areas were missing. It is also possible that these nodes required more energy due to some kind of distribution inefficiency.

Annual average heating demand varies considerably depending on the year of construction of a building. For buildings built after 1980, the heating demand was about 97–98 kWh/m$^2$ in 2004, while for those built before 1980 heating demand was from 120–133 kWh/m$^2$ per year [40]. For the sampled buildings with a documented year of construction, the average heating demand for buildings constructed before 1980 was about 246.46 kWh/m$^2$ per year, with these buildings accounting for 98,838 m$^2$ of the heated floor area, as shown in Table 1. There is, therefore, great potential—amounting to an improvement of about 13,487 MWh per year—for these buildings built before 1980 to improve their energy performance through renovations, such as increasing the thermal insulation of the walls/roofs or upgrading windows and heating radiators. The rest of the buildings in the study case accounted for 132,912 m$^2$ of the heated floor area, with a heating demand of about 296–297 kWh/m$^2$. There appeared to be no significant difference between the data for the buildings built after 1980 and those with unclassified years of construction but, due to an even higher heating demand, they still offer great potential for energy saving, around 24,917 MWh per year.

**Table 1.** Comparison between heating demand in the studied case and the average data for Sweden.

| Year/Case | Heating Demand | Area | Potential Savings |
|---|---|---|---|
| 2018, case study, 1980> | 296.90 kWh/m$^2$ | 11,315 m$^2$ | 2114 MWh/m$^2$ |
| 2018, case study, <1980 | 246.46 kWh/m$^2$ | 98,838 m$^2$ | 13,487 MWh/m$^2$ |
| 2018, case study, N/A | 297.53 kWh/m$^2$ | 121,597 m$^2$ | 22,803 MWh/m$^2$ |
| 2018, Swedish building code, [36] | 110 kWh/m$^2$ | Historical heating demands are shown in the left columns according to building code and practice. In comparison to the studied case, great potential savings in heating can be observed. | |
| 2014, Swedish practical average, [39] | 154 kWh/m$^2$ | | |
| 2010, Gävleborg practical average, [39] | 184 kWh/m$^2$ | | |
| 2004, Pallardó [40], 1980> | 97 to 98 kWh/m$^2$ | | |
| 2004, Pallardó [40], <1980 | 120 to 133 kWh/m$^2$ | | |

### 3.2. Spatial Data Analysis

A digital mapping method was applied in this study to compile and format the energy data into a virtual image and thus to produce a general map of energy use in Borlänge city based on the building samples, offering appropriate representations of the dedicated areas and districts.

By using a geographic information system tool—QGIS—it was possible to visualise the sample energy data on a spatial map of Borlänge [41]. Using the yearly electricity and heating demand, as measured in the unit kWh/m$^2$, as the weight factor, along with the longitudes and latitudes of the addresses, two digital maps were generated, as shown in Figures 6 and 7, for electricity use and heating demand, respectively.

These digital maps provide an interactive and scalable way of visualizing the energy use across the city, which can be used to spot abnormalities or faulty energy data points. They also provide a spatial representation of identifiable hotspots for electricity uses in high-occupancy/density areas. For district heating demands, they show hotspots with buildings mostly constructed before 1980. For instance, some of the hotspots can be easily identified as several student accommodation areas in the northwest quadrant. These highly dense buildings showed high electricity consumption since the occupants remain indoor for most learning and living activities; but, at the same time, these buildings had relatively low heating needs as the buildings are well maintained and insulated. It can be observed from these two maps that electricity use mainly depended on the occupancy density, with higher population per floor area usually resulting in higher electricity use. On the other hand, district heating demand was dependent on the building itself, with poorly-insulated buildings leading to higher heating need. As a result, electricity use and heating demand did not always appear in the same district/area since they were influenced by different parameters. This results offer clear insights for the planning of urban energy infrastructure and distribution, as well as with regard to the potential contributions from local renewable

energy source (RES) systems. For instance, more extensive electricity distribution or greater RES power generation are necessary for highly dense residential areas, while better heating should be distributed to those areas with buildings mostly constructed before 1980.

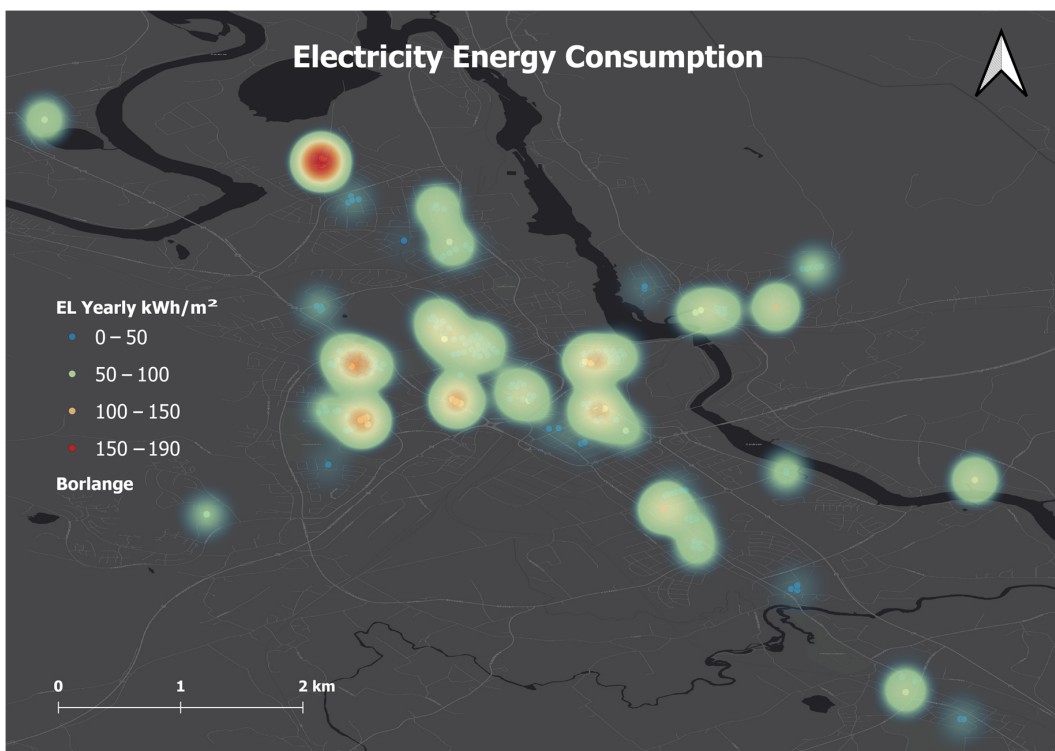

**Figure 6.** Digital mapping of electricity use in Borlänge city based on building samples.

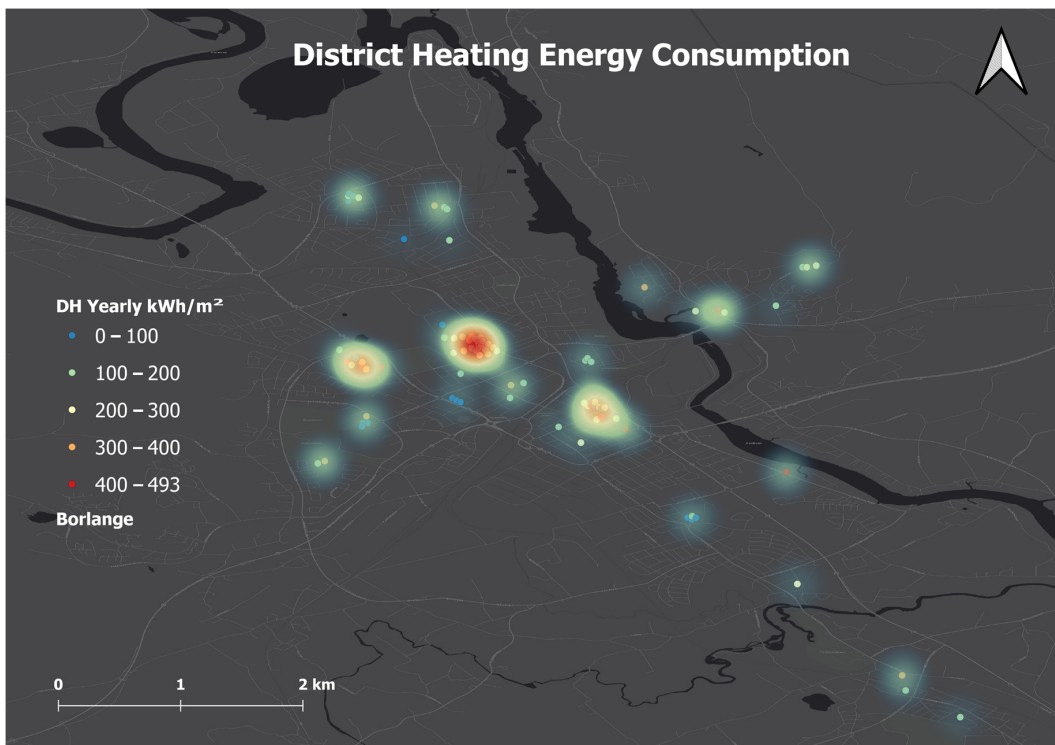

**Figure 7.** Digital mapping of heating demand in Borlänge city based on building samples.



### 3.3. Temporal Data Analysis

The yearly energy aggregation, providing a global overview of the data, was analysed in the previous section, but energy demand varies strongly depending on time. Seasonal and daily patterns have been regularly noted in the literature. In this section, we report on the analysis of monthly energy consumption. The same methodology and assumptions as before are applied, in this case at the monthly scale.

The air temperature data for the year 2018 in Borlänge indicated that there were direct correlations with the energy consumption and temperatures. During the winter months, the temperature drops below 0 °C. Afterwards, a short spring rapidly transitions into the summer season, which is accompanied by a pleasant temperature around 20 °C. A relatively smooth transition from autumn to winter occurs between September and November.

There was a slightly negative correlation between the electricity demand and the temperature at the monthly scale. The correlation was stronger for district heating because not only the median values but also the maximum values of district heating energy showed a significant decrease from April and increase from August (Figures 8 and 9).

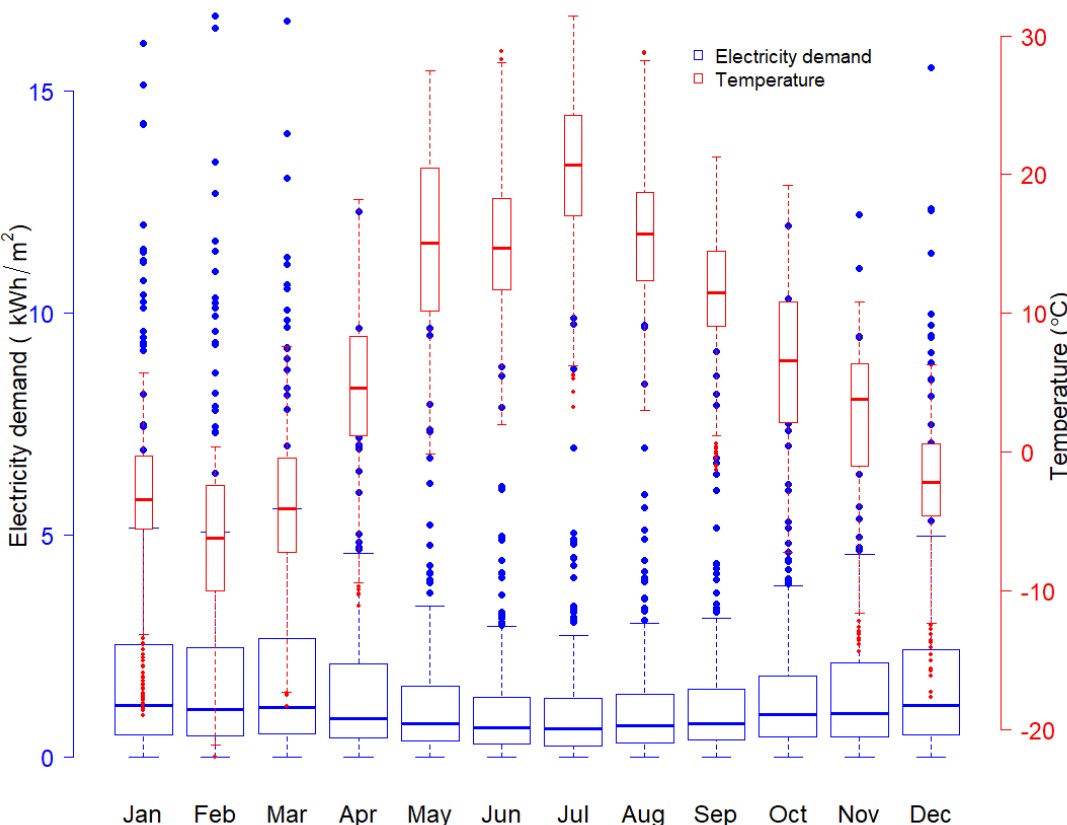

**Figure 8.** Monthly electricity demand and air temperature in Borlänge in 2018.

Further analysis of the dataset provided a better understanding of the temporal dimension, as the previous analysis was outlined based on a yearly aggregation. On the monthly temporal scale, the seasonal weather impact was more evident. Winter usually refers to December, January, February and March in Borlänge. In this period, the average electricity consumption was 2.65 kWh/m² per month. In contrast, the summer season, comprising May, June, July and August, had an average electricity consumption of 1.52 kWh/m² per month. The average consumption for the transitional seasons, spring and autumn, was 1.94 kWh/m² per month. Table 2 shows the descriptive statistics, such as the mean value, minimum and maximum monthly and yearly values for 2018 and the median and standard deviation calculations. It can also be seen that the variations of electricity demand in winter were higher than in other seasons. One possible reason might have

been that additional electricity heaters and more lighting devices were used in winter. The electricity usage varied depending on individual factors and the effect was significant in summer. Figure 8 shows the electricity demand per month in a boxplot. The high monthly electricity demand data series corresponds to laundry, parking and other facilities, with a peak monthly consumption above 15 kWh/m$^2$ per month. It can be further observed that almost all the "outliers" appear in the upper part of the boxes and that the number of "outliers" is similar for each month. This explains why the mean values are higher than the median. Thus, the influences of these "outliers" should be noticed when evaluating load distribution.

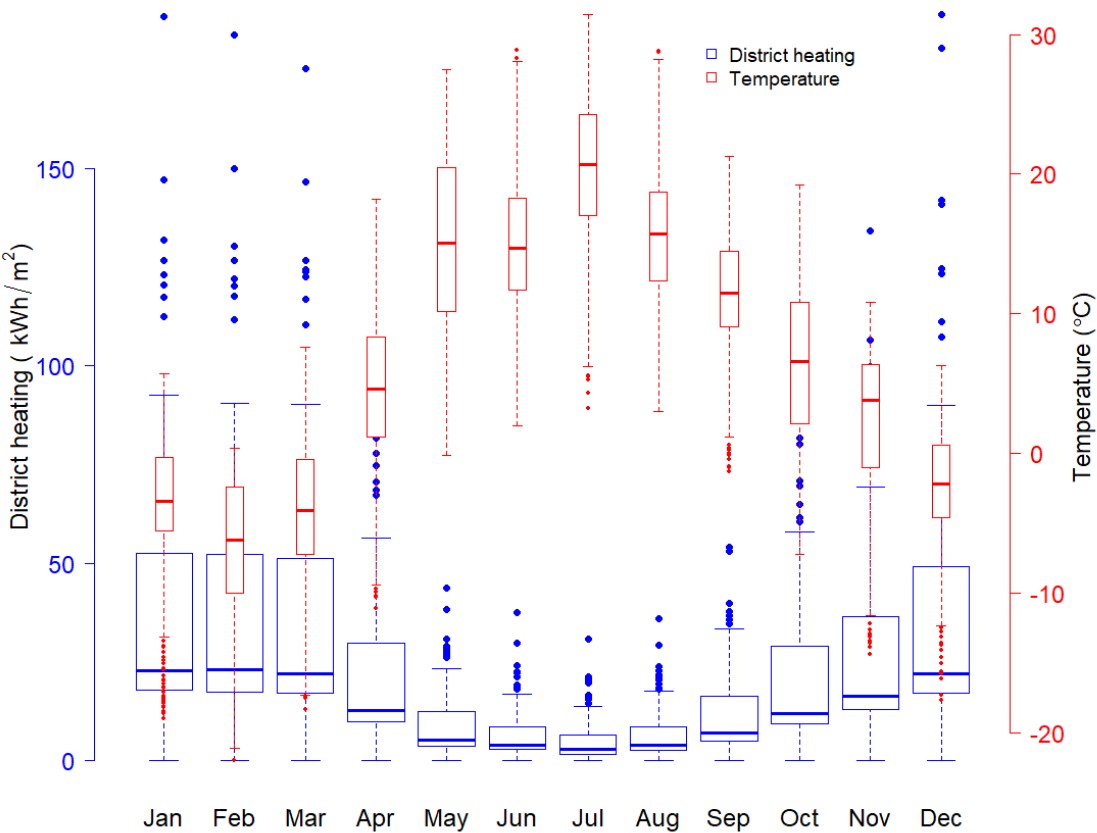

**Figure 9.** Monthly district heating demand and air temperature in Borlänge in 2018.

**Table 2.** Electricity demand per month: averages, minimum, maximum and median.

| 2018 kWh/m$^2$ | Mean | Min | Max | Median | Standard Deviation |
|---|---|---|---|---|---|
| January | 2.818 | 0.018 | 24.966 | 1.459 | 3.648 |
| February | 2.648 | 0.014 | 24.044 | 1.330 | 3.472 |
| March | 2.621 | 0.016 | 22.870 | 1.331 | 3.348 |
| April | 1.975 | 0.012 | 14.970 | 1.091 | 2.435 |
| May | 1.632 | 0.010 | 13.409 | 0.917 | 2.068 |
| June | 1.468 | 0.001 | 13.015 | 0.813 | 1.949 |
| July | 1.441 | 0.008 | 13.020 | 0.776 | 2.041 |
| August | 1.540 | 0.007 | 13.009 | 0.858 | 2.020 |
| September | 1.666 | 0.016 | 13.339 | 0.937 | 2.074 |
| October | 1.971 | 0.017 | 14.229 | 1.169 | 2.432 |
| November | 2.165 | 0.017 | 17.060 | 1.210 | 2.640 |
| December | 2.524 | 0.022 | 22.065 | 1.324 | 3.139 |
| Yearly | 24.470 | 0.163 | 189.886 | 13.175 | 29.384 |

District heating energy demand was significantly more consistent than electricity demand, as shown in Table 3 and Figure 9. For the winter months, there was an average heating energy demand of 38.8 kWh/m$^2$ per month. In contrast, summer had an average consumption of 7.27 kWh/m$^2$ per month. The transitional seasons, spring and autumn, had an average of 21.11 kWh/m$^2$ per month. The intra-difference for each winter and summer seems to be negligible while the inter-difference is obvious.

**Table 3.** District heating demand per month: averages, minimum, maximum and median.

| 2018 kWh/m$^2$ | Mean | Min | Max | Median | Standard Deviation |
|---|---|---|---|---|---|
| January | 39.232 | 0.037 | 188.436 | 22.741 | 34.686 |
| February | 39.204 | 0.037 | 183.786 | 23.068 | 34.354 |
| March | 38.448 | 0.045 | 175.248 | 21.941 | 33.809 |
| April | 22.698 | 0.023 | 99.735 | 12.767 | 20.404 |
| May | 9.603 | 0.043 | 43.763 | 5.304 | 8.959 |
| June | 7.192 | 0.043 | 37.452 | 4.083 | 7.012 |
| July | 5.300 | 0.017 | 30.831 | 2.979 | 5.586 |
| August | 6.987 | 0.033 | 36.026 | 3.876 | 6.827 |
| September | 12.082 | 0.022 | 54.051 | 7.115 | 10.980 |
| October | 21.700 | 0.048 | 100.274 | 11.978 | 19.515 |
| November | 27.981 | 0.030 | 134.077 | 16.269 | 24.970 |
| December | 38.361 | 0.038 | 188.769 | 21.958 | 36.329 |
| Yearly | 268.788 | 0.414 | 1234.521 | 149.347 | 238.991 |

Thus, comparing the electricity demand to the district heating monthly energy demand, it is observed that district heating adhered to a more homogenous pattern following the changes of the seasons. For the sampled buildings, the heating was managed by central systems. The variations were determined more by the building envelopes, physical parameters and weather conditions than by the occupant behaviours. This explains the regular pattern and the lower number of "outliers" for each month. The high heating demand values might have been due to poor insulation material or inefficient energy systems.

### 3.4. Information Map of Spatial–Temporal Energy Demand

This part of the study aimed to increase the level of detail of the spatial–temporal energy analyses with the intention of presenting a digital spatial–temporal information map of both electricity use and district heating demand. The initial data consisted of electricity and district heating monthly energy consumption for the year 2018, visualised in graphs and mapped in 2D heatmaps for electricity and district heating, respectively. We expanded the information from the energy use map for benchmarking of large-scale buildings. For this purpose, a new database structure was created by merging the spatial information from Lantmäteriet, the database with the geocoded addresses and the temporal energy demand. The basic workflow for merging these datasets is shown in Figure 10.

The process consisted of using the two databases, in the ".csv" and ".shp" formats, to create a single database with all the features from both files. For the geocoded data, the coordinates were in the standard WGS-84 format and had to be re-projected into the SWEREF-99-TM format to be consistent with the survey data. The re-projected point coordinates were then extracted and used as markers to select the polygons from the survey data. In parallel, the coordinates and areas were extracted from the survey data and used as an underlay, as shown in Figure 11. The enhanced polygons then contained the energy information.

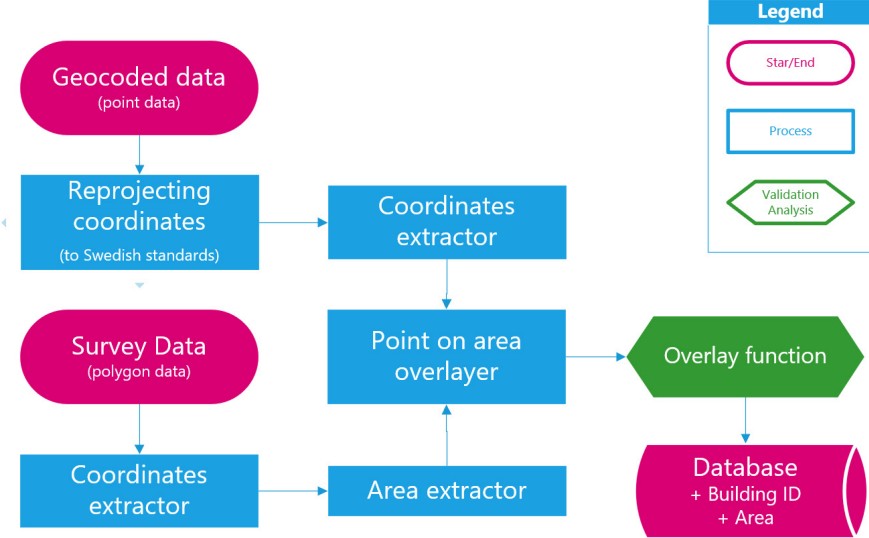

**Figure 10.** Area merger workflow.

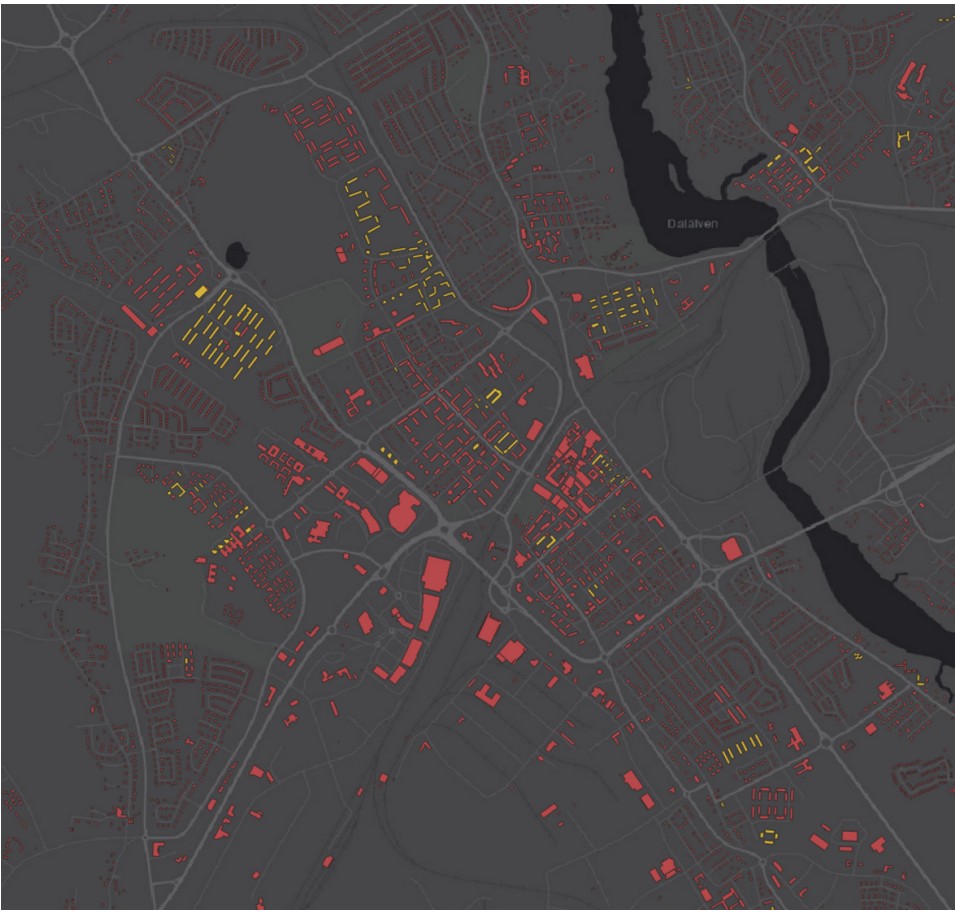

**Figure 11.** Borlänge building footprints (in red) and this study's building footprints (in yellow).

The merged database, now in the ".shp" file format, contained the areas, identification numbers and general parameters for the buildings, as well as the energy demand information per month in kWh/m$^2$, as show in Figure 12. Unnecessary data were filtered out of the dataset.

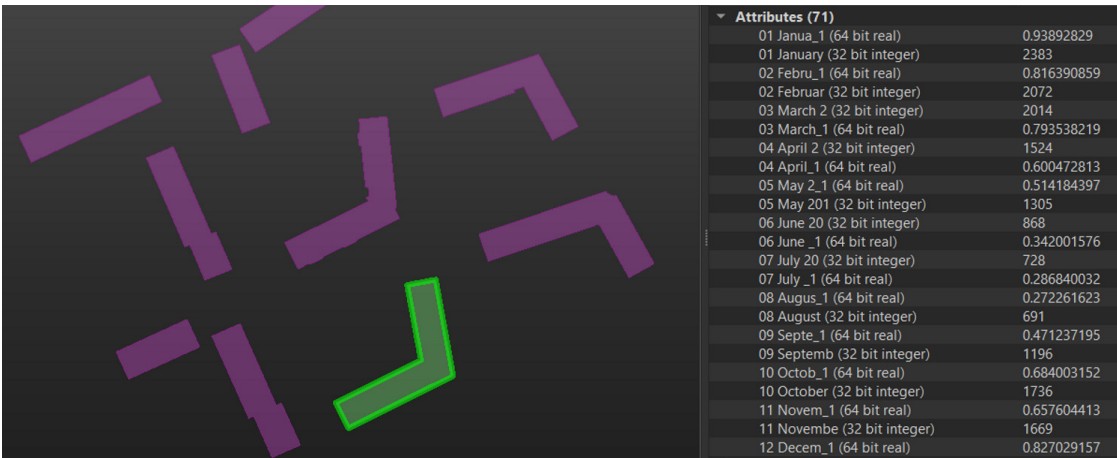

**Figure 12.** This study's building footprints with energy use information.

### 3.5. D Visualization Methodology for the Integrated Spatial–Temporal Energy Demand

It was furthermore possible to visualise the integrated spatial–temporal energy demand information in a 3D map. This activity was achieved with a novel digitalization approach. Three data files were required to generate the 3D model, obtained from Lantmäteriet and from the new database shapefile: (i) laser imaging, detection and ranging (LIDAR) data, used for measuring distances by illuminating the target with laser light and measuring the reflection with a sensor; (ii) vector maps of the building properties in the "shapefile" format, which is a geospatial vector data format for geographic information system software; and (iii) an "orthophoto" file, which is an aerial photograph or satellite imagery geometrically corrected to a uniform scale.

The methodology involved a parallel data transformation of the three data files using the Feature Manipulation Engine (FME) tool. The data were deconstructed and reconstructed until the desired outcome was achieved. The building shape data could be used as a clipping mask on both the "orthophoto" and the LIDAR data, defining the building boundaries and creating a unique building identification. The "orthophoto" could be then applied to generate textures for building roofs, as well as for the terrain. The LIDAR data was used to generate the building geometries and heights and the terrain elevation.

Consequently, a 3D city model could be finally generated, as illustrated in Figure 13, with the purpose of complementing the generation of an urban modelling framework/information. The model could further include the spatial–temporal energy use information, as indicated in Section 3.4, and thus assist the master energy planning of the buildings in the city.

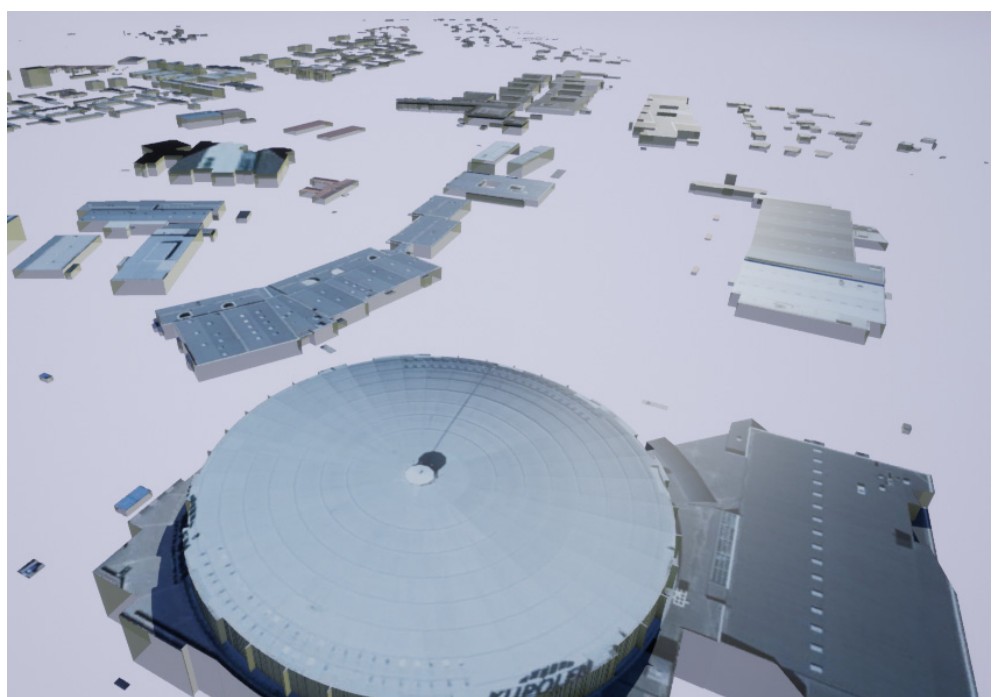

**Figure 13.** Example of 3D model visualization.

## 4. Discussion

This study aimed to establish the groundwork for further urban-scale energy exploration. The methodology developed here made it possible to pre-process and validate the data for further analyses and could be applied to other study cases. This serves to demonstrate that, even with a limited set of data, meaningful models can be created when the data are manipulated and processed properly, as shown in the geocoding process or in the 3D model creation.

Data analyses were done for different temporal and spatial scales; yearly averages showed correlations for the energy demand, year of construction and typology of the buildings, indicating the potential for future renovation opportunities. The monthly study showed a direct correlation between energy demand and weather patterns. The spatial analysis put into perspective the different zones and clusters of buildings in the urban landscape. It showed that the distribution of district heating demand was greater closer to the centre of the energy production source, while it dispersed in the outskirts; electric energy demand increased in the outskirts, especially during the winter time, explained by the use of electric heaters in the absence of district heating.

Moving from building-level energy modelling towards urban-level modelling presented many challenges. Data were often difficult to acquire and entailed many obstacles, such as privacy laws, the format in which data were stored, data accuracy and, at times, a lack of any data at all. Data pre-processing and processing can also be time consuming and highly demanding on computational power, leading to a compromise between accuracy and simplified assumptions. In this paper, a lot of attention was put into the level of detail of the data, avoiding simplification where possible. In future analyses, the dataset parameters will be submitted to machine learning models to observe the relationships and dependencies between each element, such as between weather conditions, energy demand and building efficiency. The dataset will be expanded to incorporate a large number of extra parameters, from weather data—like air temperature, humidity and irradiance information—to social data and energy data—like energy certification ratings—thus expanding our understanding of the relevance of the specific parameters.

Examining energy patterns at a yearly scale provides a general overview of high-consumption dwellings and helps in categorising them in terms of their function and efficiency. Information about potential energy saving can be obtained and comparisons can be drawn about the overall energy performance of a city in relationship to the country regulations. Examining energy patterns at a monthly scale, on the other hand, specifically shows the relationship between energy demand and seasonal changes. This level of detail makes it possible to see how a building performs under different circumstances. For instance, some buildings might perform well in the winter and in the summer but might require more energy for cooling due to high insulation or limited ventilation. Attention should be paid to the relationship between the energy demand and weather conditions at a larger scale, moving from years to months, days or even minute resolutions. In the spatial context, moving from one dimension (data points) to two dimensions (data on a plane; maps) expands the perception and interpretation of the data analysed, allowing spatial relationships to be understood better. The next step is to expand the spatial data in such a way that leads toward a 3D model, in which the data can be explored in a more direct and realistic manner.

Energy master planning (EMP), at the district and city levels, provides the possibility of untangling the challenges to the dynamics of energy needs and supply. The detailed 3D city information model is an essential digital EMP platform to engage different stakeholders in communication and thus help them to identify their roles in sustainable energy transition. In this model, buildings have shape and volume, the sun casts shadows and vegetation is present. These data explicitly convey a lot of extra information to stakeholders—energy and urban planners—allowing them to explore the data through an interactive platform.

In future work, the main focus will be on creating a 3D model incorporating a high level of detail for buildings, terrain, energy demand and other building characteristics. Visualizing large amounts of information is challenging and for this purpose a graphical user interface (GUI) must be created to enable interaction with the energy information in the 3D model, making it possible to show, hide or filter various information either with numbers or colour codes.

## 5. Conclusions

A dedicated spatial–temporal analysis of both electricity use and district heating demand in a Swedish local-city context was provided in this study using a toolkit for top-down digital mapping. The average electricity demand in the Borlänge building samples was 24.47 kWh/m$^2$, which was reasonably lower than the average value in Sweden. The mean value of heating of the building samples was 268.78 kWh/m$^2$, which was much higher than either the building code or the passive house standard. The heating use in Borlänge city remained at a high level when compared to the closest regions and the average figure over the country. In particular, there was great potential for the improvement of energy performance, amounting to savings of about 13,487 MWh/year for the buildings built before 1980 and around 24,917 MWh/year for the rest of the buildings.

The digital maps provided a spatial representation of the identifiable hotspots for electricity uses in high-occupancy/density areas and for district heating needs in districts with buildings mostly constructed before 1980. Visualizing the energy use across the city also showed that there was some apparent correlation in the electricity use and heating demand hotspot locations, as, for example, in the increase of electricity use and decrease of district heating in the periphery.

Further expanding the temporal scale from yearly to monthly values made it possible to study the electricity use and the district heating pattern in relation to the changes of the seasons and temperatures over the year. As expected, heating demand was increasingly relational to the decrease of temperatures, as was the electricity demand, although at a lower intensity.

The approach to generating the information map for the spatial–temporal energy demand was finally concluded with the three datasets including spatial information from

Lantmäteriet, geocoded addresses and temporal energy demand. This method also expanded the potential to integrate energy information into city information models at a 3D level through parallel data transformation of the three data files with the Feature Manipulation Engine tool. The overall result offers clear insights for the planning of urban energy infrastructure and distributions, as well as a potential contribution for local RES implementation.

**Author Contributions:** Conceptualization, S.Q. and X.Z.; methodology, S.Q., M.H., and X.Z.; formal analysis, S.Q. and X.Z.; writing—original draft preparation, S.Q., P.H.; writing—review and editing, P.H., M.H. and X.Z.; visualization, S.Q. and M.H.; supervision, X.Z. All authors have read and agreed to the published version of the manuscript.

**Funding:** The authors would like to note the financial support from the Swedish Energy Agency (UBMEM project: 46068).

**Institutional Review Board Statement:** Not applicable.

**Informed Consent Statement:** Not applicable.

**Data Availability Statement:** The data presented in this study are available on request from the corresponding author.

**Acknowledgments:** The authors also thank Tina Lidberge for accruing data from Tunabyggen. The masters students, such as Péter Tempfli, Mohsin Raza, Anastasiia An and Mrudula Talari, are appreciated for their support.

**Conflicts of Interest:** The authors declare no conflict of interest.

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
