# Peer review of "A Top-Down Digital Mapping of Spatial-Temporal Energy Use for Municipality-Owned Buildings: A Case Study in Borlänge, Sweden"

_buildings, doi:10.3390/buildings11020072_

Round 1
Reviewer 1 Report
Thank you for very interesting case study from the city of Borlänge, Sweden.
Please, notice in Figure 4 Annual electricity demand for building samples. Legends of curves are missing.
Author Response
Thank you very much for the positive feedback.
Reviewer 2 Report
The data is interesting and gives nice approach to find out district energy use. The paper is useful to give also for practical point of view as it allows to geolocate the possible energy saving recommendations. The findings of this paper will support city policies towards low and zero carbon cities.
Author Response

(The authors gave the same response as above.)

Reviewer 3 Report
The paper presents a methodology fort the calculation of electricity use and district heating demand in spatial-temporal dimension. A digital spatial mapping of a set of buildings in the city of Borlänge, Sweden, has been realized. The abstract is clear and concise. In the introduction all the topics have been treated in a exhaustive way. Just few approaches are missed. The part on top-down and bottom up approaches doesn’t consider the European researches on the energy performance of an urban area based on the “typology approach”, such ass the International Energy Europe projects Tabula, Episcope and RePublic_ZEB. This part improve a lot the knowledge in the energy performance of buildings in urban areas. Also, the potentials of the spatial Cluster Analysis for supporting urban studies are not considered. This approach permits to understand better the that parameters can influence the evaluation of energy demand according to the building stock, thus it is useful for your research. For more see https://doi.org/10.26868/25222708.2019.210346. Also, the use of Sina Weibo POI Data for analyzing urban spatial patterns is not consider, that is particularly innovative. For more information see https://doi.org/10.3390/su13020647. The structure of the paper is really clear. Particularly the Flowcharts explicate very well the methodology of the research. Regarding the energy data, I suggest you to summarize the data in rows 310-318 in a table to improve their legibility. Also, a comparison between these energy data and the average data of Sweden could help to understand better the energy performances. Figure 8 wants to show the correlation between air temperature and energy consumptions. I suggest to cut this figure, ant to add these data to figure 9 and 10. In this way, the correlation is more clear. Conclusion are not enough. More important findings are inserted in the paper, I.e. from row 340 to the end of the section.
Author Response

(The authors gave the same response as above.)

Round 2
Reviewer 2 Report
Good improvments
Reviewer 3 Report
-